# Hydrogen Embrittlement of CrCoNi Medium-Entropy Alloy with Millimeter-Scale Grain Size: An In Situ Hydrogen Charging Study

**DOI:** 10.3390/e25040673

**Published:** 2023-04-18

**Authors:** Shaohua Yan, Xipei He, Zhongyin Zhu

**Affiliations:** 1College of Physics and Optoelectronic Engineering, Shenzhen University, Shenzhen 518060, China; yanshaohua6@163.com (S.Y.);; 2Engineering Training Center, Southwest Jiaotong University, Chengdu 610031, China

**Keywords:** CrCoNi medium entropy alloy, hydrogen embrittlement, in situ hydrogen charging, deformation twins, EBSD

## Abstract

In this study, we examined the effect of charging current density on the hydrogen embrittlement (HE) of MEA and the associated HE mechanisms using electron backscattered diffraction (EBSD). Results show that MEA is susceptible to HE, but is stronger than as-rolled and 3D-printed Cantor alloy and stainless steel. The HE susceptibility of MEA decreases with increasing current density. Ductile fracture with transgranular dimples switches to intergranular brittle fracture with clear slip bands in the interior of grains. EBSD results uncovered that hydrogen facilitates localized slips and deformation twins. Hydrogen-enhanced localized plasticity and hydrogen decohesion are the possible HE mechanisms.

## 1. Introduction

High-entropy alloys (HEAs), which consist of five or more multiprincipal elements with equi- or nonequiatomic concentrations of each element, have received extensive attention since 2004 [1,2,3,4,5,6,7,8,9]. Among the HEAs, the CoCrFeMnNi HEA (also called Cantor alloy) a presents face-centered cubic (FCC) microstructure [10], and shows excellent room temperature fracture resistance and a sound combination of strength and ductility [11]. These mechanical properties are even better at cryogenic temperatures due to the twining-induced plasticity [12]. These excellent mechanical properties suggest that the Cantor alloy is a promising alloy for use in harsh environments.

In real industrial applications such as power plants and marine structures, Cantor alloy structures may be subjected to HE, which can cause catastrophic failure via brittle fracture. To improve the HE resistance of the Cantor alloy, several studies have been performed via pre- or in situ hydrogen charging tests. Luo et al. [13] investigated the HE of Cantor alloys under in situ charging hydrogen conditions with a strain rate of 10^−4^ s^−1^. Surprisingly, they found that hydrogen simultaneously increased both the strength and ductility; they argued that the hydrogen reduced the stacking fault energy and promoted the deformation nanotwins which enhanced the alloy’s work-hardening ability. In another paper by Luo et al. [14], the strength and ductility at a cryogenic temperature were not influenced by the presence of hydrogen. The beneficial effect of hydrogen in the research of Luo et al. may be due to the small concentration of hydrogen (~8 mass ppm) and relatively higher strain rate (10^−4^ s^−1^). HE may not occur in such experimental conditions. Zhao et al. [15] increased the hydrogen content to 77 mass ppm via gaseous and electrochemical methods, and found that the Cantor alloy’s ductility was slightly reduced by 5%, but its fracture mode was ductile. Ichii et al. [16] further increased the hydrogen to 113 mass ppm via gaseous method, and deformed the pre-charged specimens under the strain rate of 10^−2^ s^−1^ and 10^−4^ s^−1^. They reported that the Cantor alloy presented HE with intergranular cracks. The HE was more obvious when the concentration of hydrogen reached 147 mass ppm and the strain rate was 10^−6^ s^−1^, as reported by Nygren et al. [17]. Furthermore, several recent papers also reported that the HE of the Cantor alloy happened under certain experimental conditions [18,19,20,21]. Up until now, it can be concluded that the Cantor alloy is not completely resistant to HE if the concentration of hydrogen is high enough or the strain rate is relatively low.

Moreover, microstructures can also affect the HE of HEAs. The interstitial elements such as C and N benefit the mechanical properties of HEAs [22,23], but are detrimental to their HE resistance [24,25]. Because these interstitial elements can change the stacking fault energy, hydrogen diffusivity, and formation of particles (e.g., nano-carbides), these microstructural changes lead to the deterioration of HE resistance. Other microstructural changes in grain size, dislocation density, and phases can also change HE resistance of HEAs, as reported in Refs. [18,26,27].

To date, most studies focus on the HE of Cantor alloys. However, CrCoNi MEA, one of the Cantor alloy derivatives, exhibits the best combination of strength and ductility [28,29,30], fracture and impact toughness [31,32], and sound welded joints [1], which are attributed to a lower stacking fault energy (inducing deformation twining [30]) and short-range-ordered structures [33,34]. These properties indicate that CrCoNi MEA has great potential in industrial applications. Nevertheless, investigation regarding its HE resistance is limited compared to that of the Cantor alloy and its derivatives. Soundararajan et al. [35] reported that 1 GPa-strong CoCrNi MEA had a higher HE resistance than the conventional alloys; its strength was enhanced with the presence of hydrogen, but its ductility was slightly reduced. The study by Soundararajan et al. was conducted under hydrogen pre-charging conditions and a strain rate of 10^−4^ s^−1^. As the above literature reviews, the HE is affected by experimental conditions and microstructures. However, the effect of in situ hydrogen charging and microstructure on the HE of CoCrNi MEA is still unknown.

In this paper, we investigated the HE behavior of CrCoNi MEA via slow strain rate (1 × 10^−5^ s^−1^) testing under in situ hydrogen charging conditions. Different charging current densities were utilized to induce different contents of hydrogen into the MEA. The evolution of microstructures, fracture mode, and HE cracks were characterized by scanning electron microscope (SEM) and EBSD. The HE mechanisms are discussed as well.

## 2. Experimental Methods

The equiatomic CrCoNi MEAs were fabricated from pure Ni, Co, and Cr powders in a vacuum atmosphere. The NiCoCr ingot was remelted three more times to ensure chemical homogenization. The chemical composition of the MEA was examined by energy dispersive X-ray spectroscopy (EDS), and is shown in Table 1. X-ray diffraction was used to uncover the phases of the MEA; the parameters for the testing were: Cu target, 40 kV/80 mA, scanning speed 10 °/min, 2θ = 20° − 120°.

The ingot was cast into a mold with dimensions of 100 mm × 100 mm × 3 mm (thickness). We cut several samples using electrical discharging machining (EDM) for subsequent microstructural observations. The samples were then grinded and polished with the same recipe as in Refs. [8,36,37,38]. Briefly, these samples were grinded using sandpaper with grades of #400, 600#, 800#, and #1500, then polished using diamond paste (6 μm and 1 μm), and finally polished using colloidal silica.

An etching solution (3.75 g CuSO_4_, 1.8 mL H_2_SO_4_, and 25 mL HCl) was applied onto one sample to reveal the microstructure, which can be observed in an optical microscope. Another sample was used for EBSD testing in a FEI SEM.

Dog-bone-shaped samples, whose dimensions are shown in Figure 1a, were fricated via EDM. Before the slow strain rate testing (SSRT), the samples were carefully grinded using fine SiC sandpaper. SSRT was conducted at a typical slow strain rate of 1 × 10^−5^ s^−1^ [39]. During the SSRT, in situ electrochemical hydrogen charging was used to introduce hydrogen into the MEA, as shown in Figure 1b. The solution for in situ charging is 0.5 mol/L H_2_SO_4_ + 2 g/L CH_4_N_2_S. The MEA and platinum worked as cathode and anode, respectively. Different charging current densities were utilized to investigate its influence on the HE of the MEA.

After SSRT, the cross-sectional morphology of the fractured sample was observed in SEM to verify the fracture type. EBSD was used to observe the deformed microstructure close to the fracture points and cracks. The EBSD results were analyzed using Channel 5 software to obtain the inverse pole figure (IPF) map, grain boundary (GB) and twin boundary (TB) map, and kernel average misorientation (KAM) map. EDS was used to detect the elements of particles in alloy.

## 3. Results

### 3.1. Microstructure of the CrCoNi MEA

An overview of the initial microstructure of the CrCoNi MEA in its cast status is shown in Figure 2a. Its grains with embedded dendrites were at millimeter scales, and the EDS maps (Figure 2b) indicated that the Ni, Co, and Cr elements were evenly distributed. To uncover more information on the microstructure, EBSD testing was conducted. The IPF (Figure 2c) illustrated coarse grains, a finding that is the same as the optical observation (Figure 2a), and no twins were found in the initial microstructure. The phase map (Figure 2d) and XRD result (Figure 2e) verified the CrCoNi MEA having a single-phase face-centered cubic (FCC) microstructure.

### 3.2. Tensile Properties

Figure 3a shows the representative engineering stress–strain curves of the MEA with and without hydrogen. The MEA exhibited a great combination of tensile strength (650 MPa) and ductility (140% elongation) in air. Both the tensile strength and ductility were severely degraded even when the charging current density was only 0.069 mA/cm^2^; the ductility was especially deteriorated by around 57% at this small current density. This finding suggests that CrCoNi MEA cannot beat HE under in situ charging circumstances. When the charging current density was further increased, the strength and ductility were further reduced as well, as demonstrated in Figure 3b,c. However, the tendency, strength, and ductility decreasing with increasing current density slowed when the current density was higher than 166.67 mA/cm^2^. This may be caused by the content of hydrogen via diffusion into the MEA. At a lower current density, hydrogen can diffuse into MEAs continuously, which worsens the mechanical properties, as seen in Figure 3. However, the diffusion process becomes slow after a critical current density since the content of hydrogen in the MEA reaches its limit; consequently, the effect of hydrogen on the mechanical properties becomes moderate.

### 3.3. Fractography

Figure 4a is an overview of the fractured surface of the sample without in situ charging; serious local area reduction can be observed from the SEM image. An enlarged view of the inner area of the fractured surface is provided in Figure 4b, in which ductile transgranular microvoids can be seen. Cuboidal particles (indicated by white arrows in Figure 4b) are evident in the voids. EDS maps (Figure 4c–f) confirm that these particles are likely to be Cr_2_O_3_. These particles have been reported in other studies [1,31], but their effect on mechanical properties remains unclear. However, as will be observed in the EBSD results, these particles play an important role in HE.

A typical morphology of a fractured sample with in situ charging is displayed in Figure 5a, which shows two distinct areas, i.e., brittle and ductile areas. Figure 5b, a zoomed image into the brittle area (square-marked in Figure 5a), shows a relative flat and brittle cleavage appearance. It seems that the material was peeled off along the grain boundaries during the testing. A high-resolution SEM image (Figure 5c) shows clear dislocation slip traces. The hydrogen atom can accelerate the dislocation velocity and promote dislocation slips in local areas, leading to cleavage fracture. Thus, the unique fracture morphology can be explained by hydrogen-enhanced localized plasticity (HELP) mechanisms. These dislocation slip traces are also found in another region (Figure 5d). The ductile region was full of microviods with Cr_2_O_3_ particles, evident in Figure 5e. The side of the fractured sample was mixed with slip traces and cleavage fracture, as seen in Figure 5f.

### 3.4. Deformed Microstructures

EBSD testing was used to analyze the deformed microstructure of the sample without charging, and results are shown in Figure 6. High-density deformation twins were found close to the fracture points, and these twins were parallel to each other, indicating that only primary twins occurred during deformation. The averaged value of KAM is 0.55° from the KAM results (Figure 6c). It is reported that the FCC-to-HCP transition in the microstructure may happen in the CrCoNi MEA if the local strain is higher than 50% [40]. In our case, the strain close to the fracture point was up to 130% (see the engineering stress–strain curves in Figure 3a), but no HCP phase was observed from the EBSD results. This may be caused by the resolution limit of EBSD testing, and the examination points that do not have this HCP microstructure.

Figure 7a is the deformed microstructure of the sample with an in situ charging current of 166.67 mA/cm^2^, which fractured at ~20% strain (see Figure 3a). The IPF map (Figure 7a) shows that the color in each grain is not uniform, a phenomenon due to the deformation-induced misorientation. Microviods (indicated by dark arrows in Figure 7a) are present on the grain boundaries. Deformation twins were observed, but not in high density, as demonstrated in Figure 7b. Despite the fact that the deformation twins were distributed around the grain boundaries, some deformation twins were seen in the center of the grains (Figure 7b). This finding is different from the case of the sample without hydrogen charging, in which the deformation twins were found around the grain boundaries (see Figure 6b). A possible explanation for this difference is that the hydrogen can reduce the stacking fault energy of the MEA and facilitate the deformation twins. For this reason, deformation twins can occur more easily, and thus primary and secondary twins are observed in Figure 7c. The averaged value of KAM in Figure 7d is 0.95, which is almost two times higher than that in the sample without hydrogen charging. Normally, KAM is an index of geometrically necessary dislocation density (GND) [38]. Considering that the sample without hydrogen underwent 140% strain while the sample with hydrogen underwent 20% strain, our findings indicate that the presence of hydrogen promoted the activity of dislocations and localized planar slips.

To investigate the hydrogen-induced cracks, a sample under 0.556 mA/cm^2^ hydrogen charging was investigated. Figure 8a shows four cracks whose propagation direction is almost perpendicular to the loading direction (white-arrow-indicated). As demonstrated in IPF (Figure 8b), our material exhibited transgranular cracks. In previous reports [35], FCC HEAs presented intergranular cracking in the presence of hydrogen. This different finding may be due to the microstructures. In our case, the grain size was large, meaning that the spreading path of the cracks was possibly not close to the grain boundaries. Consequently, the crack may propagate in a transgranular manner. However, intergranular cracking happened as well, as seen in Figure 5. The KAM had a higher value around the cracks, as evident in Figure 8c, and the average value of KAM was only 0.45, suggesting that there was limited plastic deformation around the cracks. The front of one crack (indicated by a square in Figure 8a) is illustrated in Figure 8d–f. It seems that the spreading path of the crack was influenced by the Cr_2_O_3_ particle, as demonstrated in Figure 8d. The KAM (Figure 8e) in this area was greater around the cracks, and had a value of 0.35, implying that this area was not greatly deformed. However, the TB map (Figure 8f) shows that there were TBs near the cracks, indicating that hydrogen facilitated the deformation twins in the CrCoNi MEA.

## 4. Discussion

### 4.1. HE Susceptibility of MEA

To evaluate the HE susceptibility of the CrCoNi MEA, we introduce the strength loss and ductility loss subjected to hydrogen attack. The strength loss ($\sigma_{loss}$) can be obtained by [18]:(1)$$\sigma_{loss}=\frac{\sigma_{air}-\sigma_{H}}{\sigma_{air}}\times 100\%$$
where $\sigma_{air}$ and $\sigma_{H}$ are the tensile strength in air and hydrogen, respectively. The ductility loss ($\delta_{loss}$) is expressed as [18]:(2)$$\delta_{loss}=\frac{\delta_{air}-\delta_{H}}{\delta_{air}}\times 100\%$$
where $\delta_{air}$ and $\delta_{H}$ are the uniform ductility in air and hydrogen, respectively. Using our results (Figure 3a) in Equations (1) and (2), we obtained the HE susceptibility under different charging current densities (Figure 9). Both the strength and ductility loss increase with the increasing current density. However, the ductility loss (up to 80%) is greater than the strength loss (up to 45%). This finding also indicates that ductility is more sensitive to hydrogen attack.

To compare the vulnerability of the HE in the MEA with other alloys, we also include alloys such as the Cantor alloy, stainless steel 301L (SS301L), and Fe-Mn-C twining-induced plasticity steel in Figure 9. It is noted that all these tests were performed at almost the same conditions regarding in situ charging and stain rate (10^−5^ s^−1^). As shown in Figure 9a, at the same charging current density, the strength loss is lower than that of SS301L [41], but greater than that of the Cantor alloy [18,26] and the Fe-Mn-C steel [42]. In terms of ductility loss (Figure 9b), the HE susceptibility of the MEA is lower than that of the as-rolled and 3D-printed Cantor alloy and SS301L, suggesting that the MEA has better HE resistance. It seems that twins can increase the HE resistance, as revealed by the strength loss and ductility of Fe-Mn-C in Figure 9. Therefore, inducing TBs into MEAs may be an effective method to improve MEAs’ resistance to HE.

### 4.2. Hydrogen-Assisted Deformation Mechanisms

As shown in Figure 3 and Figure 9, the ductility in terms of elongation is seriously degraded from 140% (without hydrogen) to 30% (with hydrogen), and the UTS is also reduced from 650 MPa to 400 MPa under the presence of hydrogen. These results confirm that the CrCoNi MEA is prone to HE under in situ charging and low strain rate (10^−5^ s^−1^) testing conditions. Resting on the experimental results, we draw a schematic of the deformation mechanisms with and without hydrogen. As shown in Figure 10a, the dislocation slips and deformation twins are the main deformation mechanisms of the sample without hydrogen. The TBs are parallel to each other, implying that only primary TBs occurred. Under in situ hydrogen charging, as shown in Figure 10b, hydrogen enters the MEA via diffusion process. As tensile strains accumulate, the GND in the MEA increases and provides the media for the hydrogen transportation and traps for hydrogen. Since hydrogen can increase the mobility of dislocations, localized slips occur (Figure 10b) when the hydrogen concentration and strains achieve the threshold value, which can be explained by the HELP theory. With the increase in hydrogens and loadings, cracks initiate from the trapping points and propagate towards inner of grains, leading to transgranular cracks, as shown in Figure 10b.

The slips localize around the GBs (see KAM in Figure 7) since the GBs act as the barrier of the dislocation motions, meaning that the GBs are distorted due to the dislocation–GB interactions. The motion of dislocation can carry more hydrogen atoms into the GBs (Figure 10b). The accumulation of H in the GBs promotes the formation of microvoids (see Figure 7a and Figure 10b), and leads to intergranular cracks as the tensile strain increases.

Besides the HE-induced cracks, deformation twins are also affected by H. As shown in Figure 8f, deformation twins can be seen near the cracks despite no occurrence of large plastic deformation. It has been reported that hydrogen reduces the stacking fault energy, which benefits the formation of twins [24]. Therefore, based on the above discussion, we can conclude that the dissolved hydrogen has two effects on the deformation behavior of MEA: the detrimental one is that hydrogen makes the motion of dislocations faster and results in localized slips according to HELP, and leads to intergranular cracks via the microvoid formation on the GBs; the beneficial one is that deformation twins, which could enhance the mechanical properties of MEA, can be more easily formed as the stacking fault energy is reduced. Since our MEA exhibits HE, it is likely that the detrimental effect of hydrogen is greater than the beneficial one.

### 4.3. Possible Factors Influencing HE of MEA

The results from pre-charged hydrogen indicate that the FCC HEA/MEA has a strong HE resistance, and can even become stronger with the presence of hydrogen atoms; results from our in situ charging experiments imply that FCC HEA/MEA is prone to HE. Notably, our results (Figure 3 and Figure 9) show that the CrCoNi MEA, despite the mm-scale grain size, experiences serious hydrogen embrittlement. These contrary findings can be explained by the following reasons:Experimental set-ups (pre-charged or in situ charged hydrogen):

HE resistance can be strongly affected by two factors, namely, the concentration of diffusive hydrogen and the depth of the hydrogen-affected zone. For the pre-charged conditions, since the diffusion coefficient of hydrogen in HEA/MEAs is constant, the amount of hydrogen entering the materials rests on the charging time. HE can be observed in HEAM/MEAs only if the concentration of hydrogen reaches a threshold value [16,20]. In fact, the reduction in tensile strength and ductility is more obvious under in situ charging conditions than pre-charged conditions, as indicated by Ref. [43]. Even if there is large amount of hydrogen in the alloys, the affected zone may be only a few tens of micrometers [19], thus, the HE is not obvious. In our case, the in situ charging experiments produced a large hydrogen-affected zone via dislocation motions and a higher concentration of hydrogen, which, consequently, leads to significant HE.
2.Strain rate:

Strain rate plays a key role in HE as well as the experimental set-ups. Normally, a low strain rate allows more hydrogen to dissolve into specific microstructures (e.g., grain boundaries or dislocation traps), and provides more time for dislocation–hydrogen interaction to enhance HELP. Furthermore, the strain rate promotes the HE cracking behaviors in terms of initiation and propagation. Therefore, HE becomes more obvious at a slower strain rate, a trend that has been confirmed in the literature. For example, Hao et al. [44] studied the effect of strain rate on the HE of a high-Mn steel and found that the HE susceptibility increased with a decrease in the strain rate. Tuĝluca et al. [45] claimed that lowering the strain rate led to the degradation of mechanical properties (work-hardening rate, elongation, and tensile strength) with the presence of hydrogen. Ichii et al. [16] investigated the effect of strain rate (10^−4^~10^−2^ s^−1^) on the mechanical behavior of pre-charged FeMnNiCoCr alloys and found that HE resistance decreased with lower strain rate. In the present case, the strain rate (1 × 10^−5^ s^−1^) was lower than in most studies (≥10^−4^ s^−1^), leading to more hydrogen diffusion into the alloys, more time for the dislocation–hydrogen interactions, and much higher localized slips both in the interior of the grains and close to the GBs. Thus, HE is more obvious in our case.
3.Microstructures:

Microstructures such as the interstitial elements C and N can alter the HE susceptibility of HEA/MEAs. Luo et al. [24] found that the tensile ductility of the Cantor alloy with 0.5 at.% C was significantly reduced when subjected to hydrogen charging, because the nanocarbides were preferable sites for crack initiation. Astafurova et al. [25] also compared the HE behavior in FeMnNiCoCr and FeMnNiCoCrN, and found that N-alloyed HEA had a higher sensitivity to HE. They claimed that the N increased the diffusion activity via increasing the lattice parameters and distortion in the HEA. In our case, the microstructure was grain size and the presence of Cr_2_O_3_, as seen in Figure 2 and Figure 4. Increasing the grain size can improve the HE resistance of HE, as indicated by Fu et al. [18]. Indeed, refined grain size would absorb more hydrogens since there are more H traps in grain boundaries, so HE sensitivity is higher in alloys with a smaller grain size. Based on this analysis, our MEA with a mm-scale grain size is expected to have strong HE resistance; contrarily, it shows high HE sensitivity. This unexpected trend, a smaller grain size having higher HE susceptivity, is also reported in high-N stainless steel [46]. It is known that hydrogen can be transported to GBs via the motion of dislocations, leading to hydrogen accumulation near the GBs. In our case, the coarsened grain can barely impede the motion of the dislocations, which facilitates the transportation of hydrogen into the GBs via dislocation motion. Such increase in hydrogen near the GBs decreases the cohesive energy of the GBs, and finally results in intergranular cracks according to the hydrogen-enhanced decohesion (HEDE) mechanism. Another factor of microstructures that influences the HE in our case is the Cr_2_O_3_ particle, which promotes the initiation of HE cracks, as demonstrated in Figure 8. A large number of the Cr_2_O_3_ particles deteriorates the tensile ductility, and dissolving these particles via heat treatment benefits the resistance to HE [27].

## 5. Conclusions

The effect of current density on the HE of a CrCoNi MEA was investigated via an in situ hydrogen charging method. The deformed microstructure and HE cracks were characterized by SEM and EBSD. Based on the obtained results and analysis, we can draw the following conclusions:

(1) At the strain rate of 1 × 10^−5^ s^−1^, CrCoNi MEA is susceptible to HE under in situ hydrogen charging conditions, and the HE susceptibility of the MEA increases with the increasing current density; however, the HE resistance of the MEA is better than that of the as-rolled and 3D-printed Cantor alloy and SUS301L. This finding further confirms the promising industrial application of CrCoNi MEA.

(2) Both intergranular and transgranular HE cracks were observed with hydrogen, with clear slip traces present in the flat interior of grains.

(3) Hydrogen facilitates deformation twins, i.e., only primary deformation twins occur without hydrogen, while primary and secondary deformation twins happen with hydrogen, and deformation twins can also be formed at the crack front where no large plastic deformation occurred; hydrogen also promotes dislocation motion and localized slip in the grain interiors and boundaries, which leads to HE crack initiation.

(4) Hydrogen-enhanced localized plasticity and hydrogen decohesion are the possible HE mechanisms of the CrCoNi MEA.

## Figures and Tables

**Figure 1 entropy-25-00673-f001:**
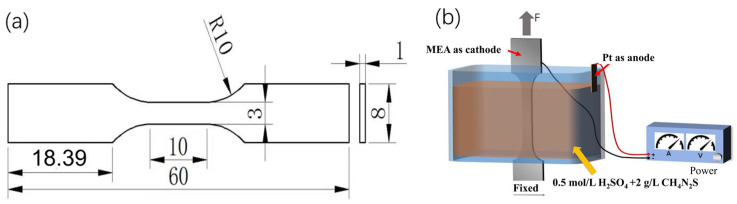
(**a**) The dimensions of the sample for SSRT, (**b**) schematic for in situ hydrogen charging during SSRT testing.

**Figure 2 entropy-25-00673-f002:**
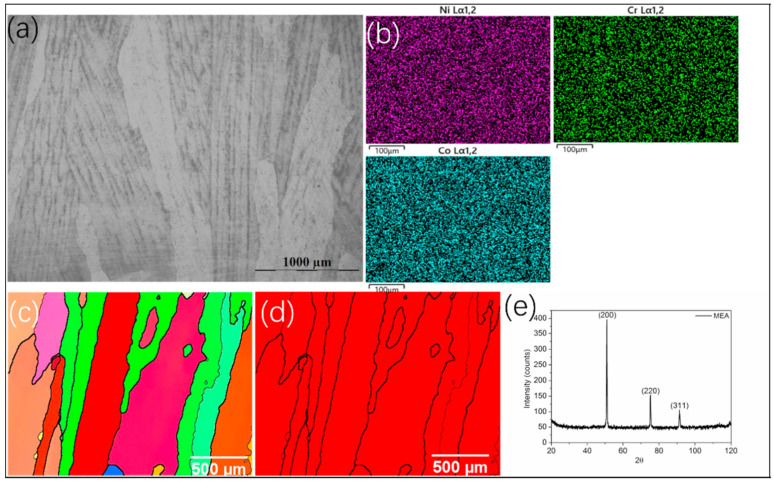
Microstructure of MEA (taken from ref. [1]), (**a**) optical image showing dendrites and grains, (**b**) EDS mapping showing Ni, Co, and Cr distributions, (**c**) IPF map, (**d**) phase map, (**e**) XRD results.

**Figure 3 entropy-25-00673-f003:**
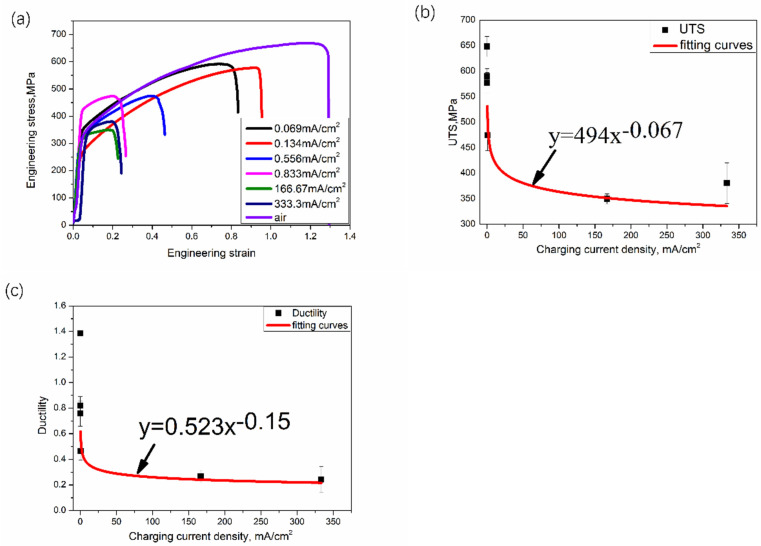
Results of SSRT, (**a**) engineering stress–strain curves obtained from SSRT under different charging current densities, (**b**) plot of UTS charging current density, (**c**) plot of ductility charging current density.

**Figure 4 entropy-25-00673-f004:**
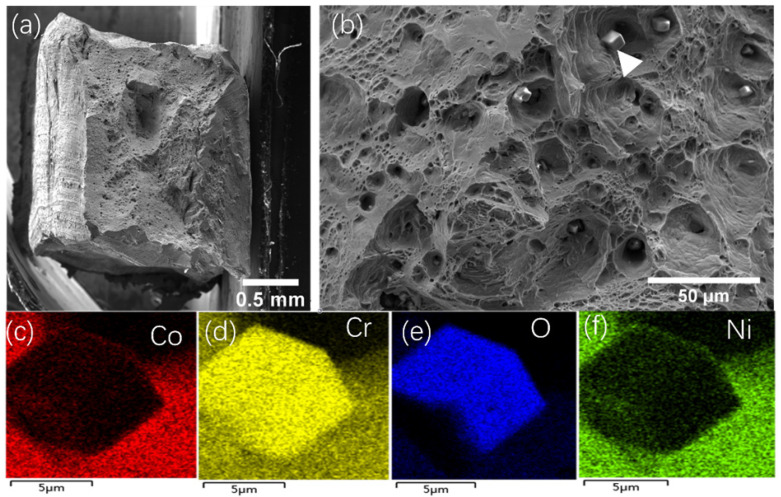
Fractography of MEA deformed in air, (**a**) overview of fractured surface, (**b**) enlarged view of the center of (**a**), (**c**–**f**) are the EDS mapping of the particle (white arrow indicated in (**b**)).

**Figure 5 entropy-25-00673-f005:**
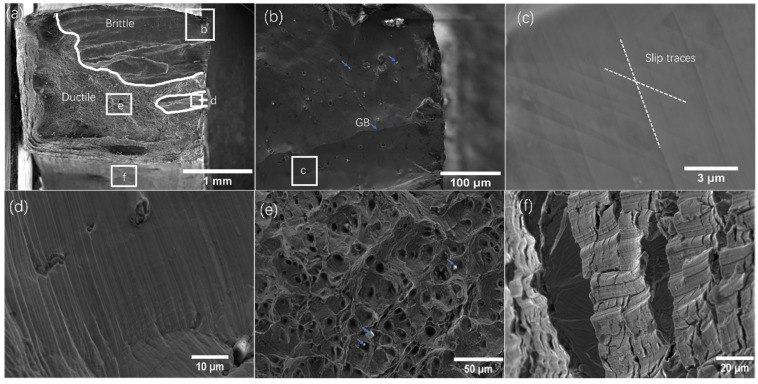
Fractography of MEA deformed with hydrogen, (**a**) overview of the fractured surface with clear brittle and ductile regions, (**b**) intergranular fracture in the brittle regions showing Cr_2_O_3_ particles (blue arrows indicated) and fracture along GB, (**c**) slip traces in one brittle region, (**d**) slip traces in another brittle region, (**e**) dimples and Cr_2_O_3_ (blue arrow indicated), (**f**) side surface showing extensive slips and brittle fracture.

**Figure 6 entropy-25-00673-f006:**
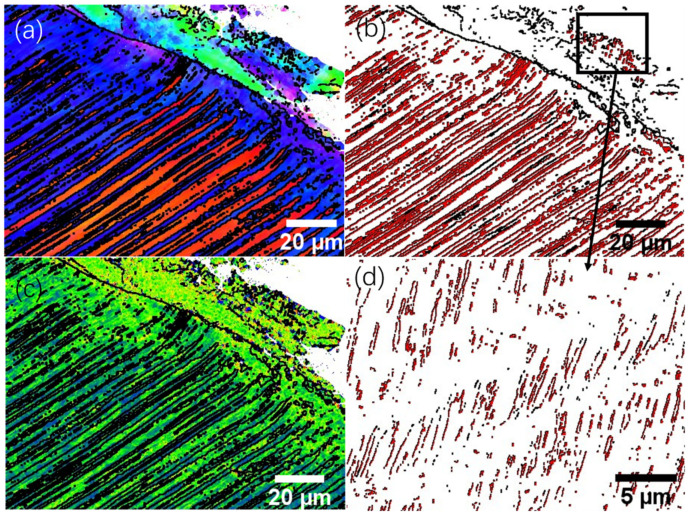
Deformed microstructure close to the fractured surface without hydrogen. (**a**) IPF map, (**b**) GB and TB map showing large amount of deformation twins, (**c**) KAM map, (**d**) enlarged view of the twin area (square-indicated) showing only primary twins.

**Figure 7 entropy-25-00673-f007:**
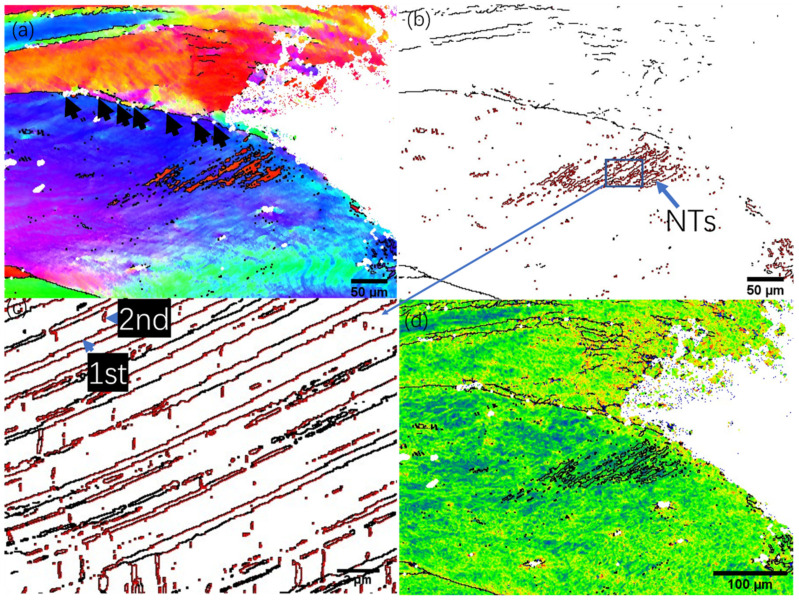
Deformed microstructure close to the fractured surface due to hydrogen. (**a**) IPF map, (**b**) GB and TB map showing deformation twins, (**c**) enlarged view of the twin area (square-indicated) showing primary and secondary twins, (**d**) KAM map.

**Figure 8 entropy-25-00673-f008:**
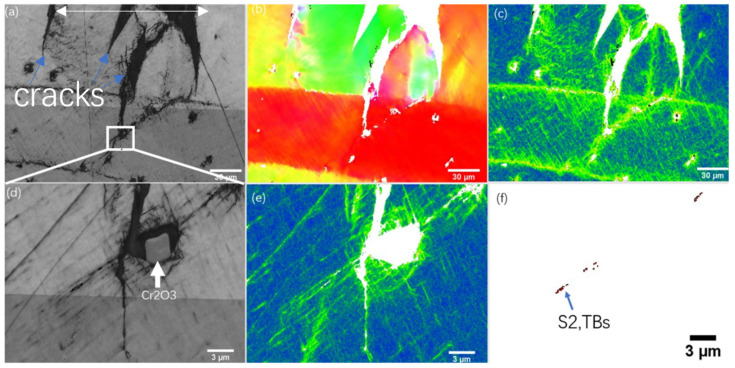
Deformed microstructure close to the HE cracks. (**a**) SEM image of a region containing cracks (blue arrows indicated), (**b**) IPF map, (**c**) KAM, (**d**) enlarged view of the square-indicated area in a, (**e**) KAM of d, (**f**) TB distribution near cracks.

**Figure 9 entropy-25-00673-f009:**
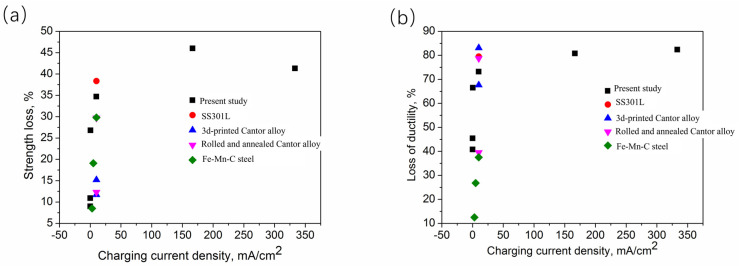
HE susceptibility in terms of strength and ductility loss of MEA under different charging current densities, (**a**) strength loss, (**b**) ductility loss. For comparison, strength and ductility loss of other alloys [18,26,41,42] under same testing conditions are also included in these two figures.

**Figure 10 entropy-25-00673-f010:**
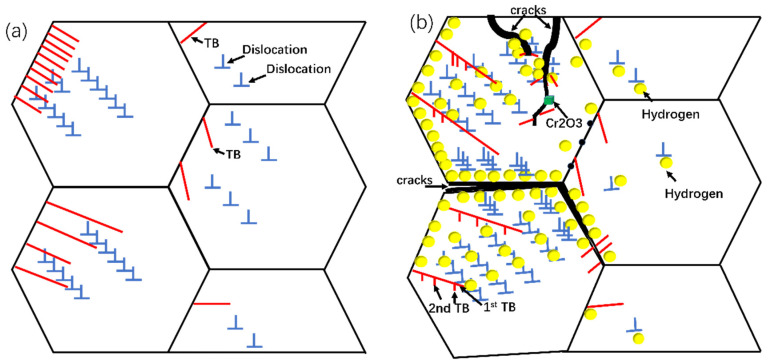
Schematics of deformation mechanisms (**a**) without hydrogen, (**b**) with hydrogen.

**Table 1 entropy-25-00673-t001:** Chemical elements of CrCoNi MEA measured by EDS.

Elements	wt. %	at. %
Ni	31.87	30.60
Co	34.84	34.84
Cr	33.29	36.08

## Data Availability

All data has been displayed in the paper.

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
