# Peer review of "Hydrogen Embrittlement of CrCoNi Medium-Entropy Alloy with Millimeter-Scale Grain Size: An In Situ Hydrogen Charging Study"

_entropy, 2023, doi:10.3390/e25040673_

Round 1

Reviewer 1 Report

Hydrogen embrittlement (HE) is a well-known phenomenon affecting the mechanical properties of structural metallic materials, but whose details concerning its mechanisms and factors are not still totally known. Although, in a great measure, the authors succeeded to put in evidence the relationship between certain microstructural and testing factors and HE for the as-cast equiatomic CoCrNi MEA (medium entropy alloy), however, some comments could be made.

a) even if the authors considered that the CoCrNi alloy have had a equiatomic chemical composition, the real chemical composition of the CoCrNi MEA should be included in the manuscript.

b) having in view the shape and dimensions of the initial sample (100 mm x100 mm x 3 mm, 3 mm heigh is much smaller in comparison with 100 mm length and 100 mm width) and relative high cooling rate (probably metallic mold at room temperature), it was not possible that the residual stresses to remain within the samples?

c) the authors found Cr2O3 particles within the CoCrNi alloy; is possible that the occurring of these particles to affect the mechanical properties of CoCrNi MEA independent of HE phenomenon; were found these particles for all the samples?

d) the authors should include in the manuscript the details regarding to the preparation procedure of samples for EBSD testing; also, details referring to the experimental conditions for XRD analysis should be included in the manuscript.

e) the elongation of the MEA without hydrogen was 140 % (air), much higher in comparison with other elongation data for this alloy from literature; how the authors explain this value?

f) both in ”3.2. Tensile properties” and ”Conclusions”, the authors stated that ” However, the tendency, strength and ductility decreasing with increasing current density, is slowing when the current density is higher than 166.67mA/cm2.” and ” the HE susceptibility of MEA decreases with increasing the current density;”; the stress-strain curves from Fig. 3 indicated an increase of the susceptibility when current density increase; indeed the rate of the susceptibility increase (decrease of mechanical properties in comparison with previous current densities) is decreasing, but susceptibility remains high; the authors should revise this part of the manuscript.

Reviewer 2 Report

In this manuscript, the authors investigated the effects of in-situ hydrogen charging on the mechanical behavior of CrCoNi medium entropy alloy. The topic of the paper is suitable for Entropy. Some revisions are to be made before the acceptance of the paper:

-          The ductility of the samples measured here are in general very large. For instance, the sample tested in air has a ductility of ~130%!! Have the authors used extensometer during the tensile tests?

-          For the current density used for hydrogen charging, there is a sudden jump in the values for 4 orders of magnitude (from 0.833 to 166.67 mA/cm2). What is the reason for such huge jump? The current density values also look rather arbitrary. How were the current density values obtained? Did the authors calculate from voltage or some other parameters?

-          On page 12, the authors mentioned “interstitial element of C and Ni.” I believe “Ni” should be “N.”

Round 2

Reviewer 1 Report

The supplementary details included in the manuscript led to an improved form of it.